# The antiandrogen enzalutamide downregulates TMPRSS2 and reduces cellular entry of SARS-CoV-2 in human lung cells

D. A. Leach[1], A. Mohr[2], E. S. Giotis [2,3], E. Cil [2], A. M. Isac[2], L. L. Yates[4], W. S. Barclay [3], R. M. Zwacka[2], C. L. Bevan [1,5✉] & G. N. Brooke [1,2,5✉]

SARS-CoV-2 attacks various organs, most destructively the lung, and cellular entry requires two host cell surface proteins: ACE2 and TMPRSS2. Downregulation of one or both of these is thus a potential therapeutic approach for COVID-19. TMPRSS2 is a known target of the androgen receptor, a ligand-activated transcription factor; androgen receptor activation increases TMPRSS2 levels in various tissues, most notably prostate. We show here that treatment with the antiandrogen enzalutamide—a well-tolerated drug widely used in advanced prostate cancer—reduces TMPRSS2 levels in human lung cells and in mouse lung. Importantly, antiandrogens significantly reduced SARS-CoV-2 entry and infection in lung cells. In support of this experimental data, analysis of existing datasets shows striking co-expression of AR and TMPRSS2, including in specific lung cell types targeted by SARS-CoV-2. Together, the data presented provides strong evidence to support clinical trials to assess the efficacy of antiandrogens as a treatment option for COVID-19.

[1] Department of Surgery and Cancer, Imperial College London, London, UK. [2] School of Life Sciences, University of Essex, Colchester, Essex, UK. [3] Department of Infectious Diseases, Imperial College London, London, UK. [4] National Heart and Lung Institute, Imperial College London, London, UK. [5]These authors contributed equally: C. L. Bevan, G. N. Brooke. ✉email: charlotte.bevan@imperial.ac.uk; gbrooke@essex.ac.uk

Severe acute respiratory syndrome coronavirus 2 (SARS-CoV-2), the cause of the COVID-19 pandemic, is a positive-sense single-stranded RNA coronavirus highly related to SARS-CoV, which caused the 2002/2003 SARS pandemic[1,2]. Like SARS, COVID-19 primarily affects the respiratory system (although other organs can also be affected): symptoms are mild in some, but in others the infection can result in pneumonia, acute respiratory distress syndrome and death[3]. Risk factors associated with poor prognosis include age, diabetes and cardiovascular disease[4]. It has also been shown that gender is a prognostic factor, with approximately 60–70% of deaths being in men[5,6], suggesting that sex steroid hormones may be a contributing factor to the severity of the disease. In further support of this, recent studies have shown that men with male pattern hair loss (caused by elevated androgen signalling[7]) are at higher risk of suffering more severe COVID-19 symptoms[8,9].

Coronaviruses have structural (spike, nucleocapsid, membrane and envelope) and non-structural (e.g., the proteases nsp3 and nsp5) proteins[10]. Cellular entry of SARS-CoV-2 requires host proteins expressed on the epithelial cell surface, most essential are transmembrane serine protease 2 (TMPRSS2) and angiotensin-converting enzyme 2 (ACE2)[11–13]. TMPRSS2 cleaves and primes the viral Spike (or S) protein; this facilitates fusion of the viral and host membranes[14–18]. Cellular entry is facilitated by ACE2, a terminal carboxypeptidase and type I transmembrane glycoprotein[19]. Thus, viral entry may be prevented or slowed by inhibition of ACE2 and/or TMPRSS2. TMPRSS2 is an attractive target as knockout of this protein causes no overt detrimental phenotype[20], whereas ACE2 downregulation is associated with increased severity of SARS-induced lung injury[21]. Furthermore, TMPRSS2 expression levels have been shown to be associated with disease severity in mouse models of coronavirus infection[22] and its inhibition was recently shown to inhibit SARS2-S-driven entry in lung cells[13].

Multiple studies have shown that TMPRSS2 is an androgen receptor (AR) target gene in prostate cancer cells[23–25]. The AR (NR3C4, nuclear receptor subfamily 3, group C, gene 4) is a nuclear receptor and member of the steroid receptor family. It is a transcription factor activated by binding of ligand (e.g., dihydrotestosterone (DHT)), upon which it translocates from the cytoplasm to the nucleus where it binds to regulatory regions of target genes as a homodimer. The active receptor then promotes gene transcription, facilitated by pioneer factors (e.g., JUN and FOXA1), and recruitment of accessory proteins and the basal transcriptional machinery[26]. In the case of prostate cancer, active AR promotes tumour growth and so treatment options for prostate cancer often target this signalling axis, through the use of androgen deprivation and hormonal therapies such as antiandrogens[27,28]. Antiandrogens (e.g., bicalutamide (BIC) and enzalutamide (ENZA)) interact with the AR in the ligand binding pocket and hold the receptor in an inactive conformation, unable to form an active transcriptional complex, and thus inhibit its activity[29]. Importantly, previous studies have demonstrated that the AR is expressed in the lung[30] and studies using mice have confirmed that AR is functional in this organ[31,32]. In corroboration, in vitro studies have shown that multiple lung lines express functional AR[31,33–35]. It is therefore possible that inhibition of androgen signalling, in response to antiandrogens, will reduce TMPRSS2 expression in the lung and reduce viral entry. For this reason, antiandrogens have been proposed as a treatment option for COVID-19[36–38].

Here we investigate AR regulation of TMPRSS2 in the mouse lung and human lung cells. We show that TMPRSS2 is co-expressed with the AR and that therapeutic antiandrogens downregulate its expression in lung. Our analysis of steroid receptor and cofactor enrichment upstream of TMPRSS2 suggests that AR may promote gene expression in both lung and prostate through distinct, cell-specific cistromic control. We show that treating human lung cells with the antiandrogen ENZA reduces entry and infection by the SARS-CoV-2 virus, providing preclinical data to support the use of antiandrogens for the treatment of COVID-19.

## Results and discussion

**TMPRSS2 is an androgen and antiandrogen-regulated gene.** ACE2 and TMPRSS2 are crucial for SARS-CoV-2 entry into cells[13], and hence these proteins represent potential therapeutic targets for COVID-19. TMPRSS2 has been shown to be an AR target gene in prostate cancer[23–25] and we therefore hypothesised that the expression of TMPRSS2 could be downregulated in response to antiandrogens. To confirm this, the AR-positive prostate cancer cell line LNCaP was seeded in hormone-depleted media for 72 h and treated with DHT (10 nM) ± 10 µM of the antiandrogens BIC or ENZA for 6 h. Alterations in gene expression were quantified using qPCR. As expected, addition of androgen significantly increased TMPRSS2 expression (Fig. 1a). Importantly, the antiandrogens successfully blocked this androgen-induced upregulation, resulting in significant inhibition of TMPRSS2 expression. TMPRSS2 expression was also significantly reduced by ENZA when cells were incubated in full medium conditions (Supplementary Fig. 1a). To determine whether AR regulation of TMPRSS2 also occurs in other cell types, gene expression was investigated in the T47D breast cancer cell line (GSE62243)[39]. In agreement with the LNCaP results, TMPRSS2 was also found to be upregulated in response to DHT in this line (Fig. 1b).

**TMPRSS2 and the AR are co-expressed in the lung.** Androgen signalling is known to be important in multiple tissues/organs. To better characterise this signalling, we previously created the AR-LUC transgenic mouse in which luciferase expression is under the control of an androgen-responsive promoter, allowing for visualisation of both in vivo and ex vivo AR activity. AR signalling was found to be active in a number of tissues/organs, including the prostate, seminal vesicles, uterus and ovaries—and importantly, AR signalling was also found to be active in the lungs of male and female mice, although activity was weaker than in the reproductive organs[32]. Other studies have also demonstrated that AR signalling is active in the lung. For example, Mikkonen et al. found the AR to be predominantly expressed in type II pneumocytes and the bronchial epithelium, and microarray analysis of the murine lung demonstrated that genes involved in oxygen transport (among other pathways) are upregulated in murine lung tissue in response to androgen[31].

To investigate AR and TMPRSS2 expression in different human tissues, we interrogated the Genotype Tissue Expression (GTEx) dataset[40]. We found that AR and TMPRSS2 are co-expressed in a number of tissues, and generally, TMPRSS2 is only expressed in tissues that also show detectable levels of the AR, with the exception of the pancreas (Fig. 2a). Importantly, AR and TMPRSS2 were found to be co-expressed in the lung (highlighted red, also highlighted are prostate, breast (both co-expressing) and pancreas). Analysis of single-cell sequencing data from lung tissue[41] demonstrated that TMPRSS2, ACE2, AR and AR-associated pioneer factors (JUN and FOXA1) are co-expressed in Epithelial Subtype Ciliated and Alveolar Type 2 (AT2) cells (Fig. 2b). Similarly in a second single-cell dataset[42], AT2 cells were among the resident lung cells with the highest expression of TMPRSS2, ACE2, and AR (Fig. 2c). These cell types are targeted by SARS-CoV-2[43]. These results are in agreement with Qiao et al., who also found in analysis of

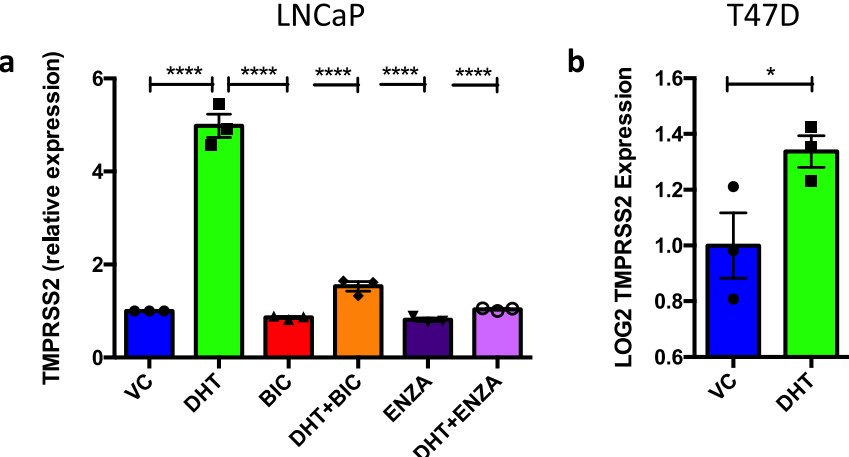

**Fig. 1 TMPRSS2 is an androgen-regulated gene in cells from different tissues. a** LNCaP cells were incubated in hormone-depleted media for 72 h and treated with ±10 nM dihydrotestosterone (DHT) ± 10 μM bicalutamide (BIC) or enzalutamide (ENZA) for 6 h. RNA was harvested, reverse transcribed and qPCR performed to quantify *TMPRSS2* expression. Mean of three independent repeats (±1 SEM). One-way ANOVA with Tukey's multiple comparison test **** Vehicle Control (VC) v DHT – $p = 8.6 \times 10^{-11}$, DHT v BIC – $p = 4.9 \times 10^{-11}$, DHT v DHT + BIC – $p = 7.3 \times 10^{-10}$, DHT v ENZA – $p = 4.0 \times 10^{-11}$, DHT v DHT + ENZA – $p = 1.0 \times 10^{-10}$. **b** Data from T47D cells (GSE62243, $N = 3$)[39] treated with 10 nM DHT were analysed for TMPRRS2 expression. One-tailed *t*-test, *$p = 0.0302$. Source data are provided as a Source Data file.

published single-cell sequencing data that the AR, TMPRSS2 and ACE2 are co-expressed in several human and mouse lung cell types, most notably AT1 and AT2 cells, and ciliated and secretory epithelial cells (bronchial cells)[44].

**TMPRSS2 expression in the lung is higher in men**. In adults, men have on average seven to eight times higher levels of circulating testosterone compared to women[45]. Men are known to have more severe symptoms following SARS-CoV-2 infection (60–70% of COVID-19-related deaths are in men[5,6]) suggesting elevated androgen signalling could be a risk factor for the disease. Furthermore, recent studies have linked male pattern hair loss (a result of elevated androgen signalling) with more severe COVID-19 symptoms[8,9]. Since *TMPRSS2* is an androgen-regulated gene, it has been proposed that elevated levels of TMPRSS2 in the lung, as a result of higher levels of androgen, might explain this gender disparity and it was therefore hypothesised that TMPRSS2 expression would be higher in male lungs compared to females. Our analysis of the GTEx dataset found no significant difference in *AR* expression levels between men and women. *TMPRSS2* expression was also found to be similar between males and females (Fig. 3), but expression in the male lung was found to be slightly and significantly higher, in agreement with a previous study[46]. This is, however, in contrast to other studies that have found no significant difference in TMPRSS2 expression in male and female lungs[37,47,48]. Furthermore, Baratchian et al. found higher levels of ACE2 in the male lung and proposed that alterations in the levels of this receptor could, at least in part, explain the gender disparity in COVID-19 severity[47]. It therefore remains unclear if gender differences in TMPRSS2 expression could explain why men suffer more severe symptoms following infection with SARS-CoV-2.

**TMPRSS2 expression is reduced by enzalutamide in lung cells**. As discussed above, the AR is expressed in human and murine lung, and shown to be active as indicated by regulation of AR-target genes in mouse lung[31], nuclear localisation of AR in (male) human lung[31,49] and activation of an androgen-responsive reporter gene in mouse lung[32]. TMPRSS2 expression has been previously shown to be androgen-regulated in the A549 cell line[31]. To replicate these findings, and to expand to other lung

lines, we seeded A549 (type II pneumocyte cell line; a cell type targeted by SARS-CoV-2[43]), H1944 (lung adenocarcinoma) and BEAS-2B (bronchial epithelial) cells in hormone-depleted media (containing serum that has been charcoal-stripped to remove any traces of hormones) for 3 days, then treated with ±10 nM DHT ± 10 μM ENZA for 24 h (Fig. 4a). In A549 and H1944, addition of DHT led to a significant increase in *TMPRSS2* expression. This induction was blocked following treatment with ENZA, although this did not reach significance. ENZA treatment in full media conditions also resulted in significant downregulation of *TMPRSS2* in A549 (Supplementary Fig. 1). Although DHT did not induce TMPRSS2 expression in BEAS-2B, ENZA did appear to reduce gene expression, suggesting that the AR could be activating gene expression in a ligand independent manner in this line. A similar trend was seen across the cell lines for another known AR target gene, *FKBP5*; this was significantly upregulated in response to DHT in A549 and H1944 and ENZA significantly reduced this induction (Fig. 4b).

To confirm AR regulation of TMPRSS2 at the protein level, A549 and H1944 cells were treated as above for 48 h (Fig. 4c) before immunoblotting. DHT increased AR levels, and ENZA prevented this ligand-mediated stabilisation of the receptor. In agreement with the qPCR data, DHT increased TMPRSS2 protein levels (approximately two-fold and five-fold, respectively) and ENZA completely blocked this induction.

**TMPRSS2 expression is reduced by enzalutamide in mouse lung**. To investigate the effects of ENZA on TMPRSS2 expression in vivo, mice were treated for 3 days with ENZA or vehicle control. Following sacrifice, lung tissue was collected and qPCR performed to quantify alterations in gene expression. While there was no significant change in *Ar* or *Ace2* expression, *Tmprss2* expression was significantly decreased after ENZA treatment ($p = 0.0367$, Fig. 5a). *Tmprss2* protein was found to be expressed in the epithelial cells and in cells of the parenchyma, as suggested by the single-cell sequencing data (Fig. 2), and intensity was visibly less in mice treated with ENZA (examples in Fig. 5b). The Ar was also found to be expressed in these cell types, with nuclear localisation indicating active Ar, and ENZA treatment resulted in a marked decrease in Ar levels in the lung (Supplementary Fig. 2). To validate these findings, gene expression data from intact mice

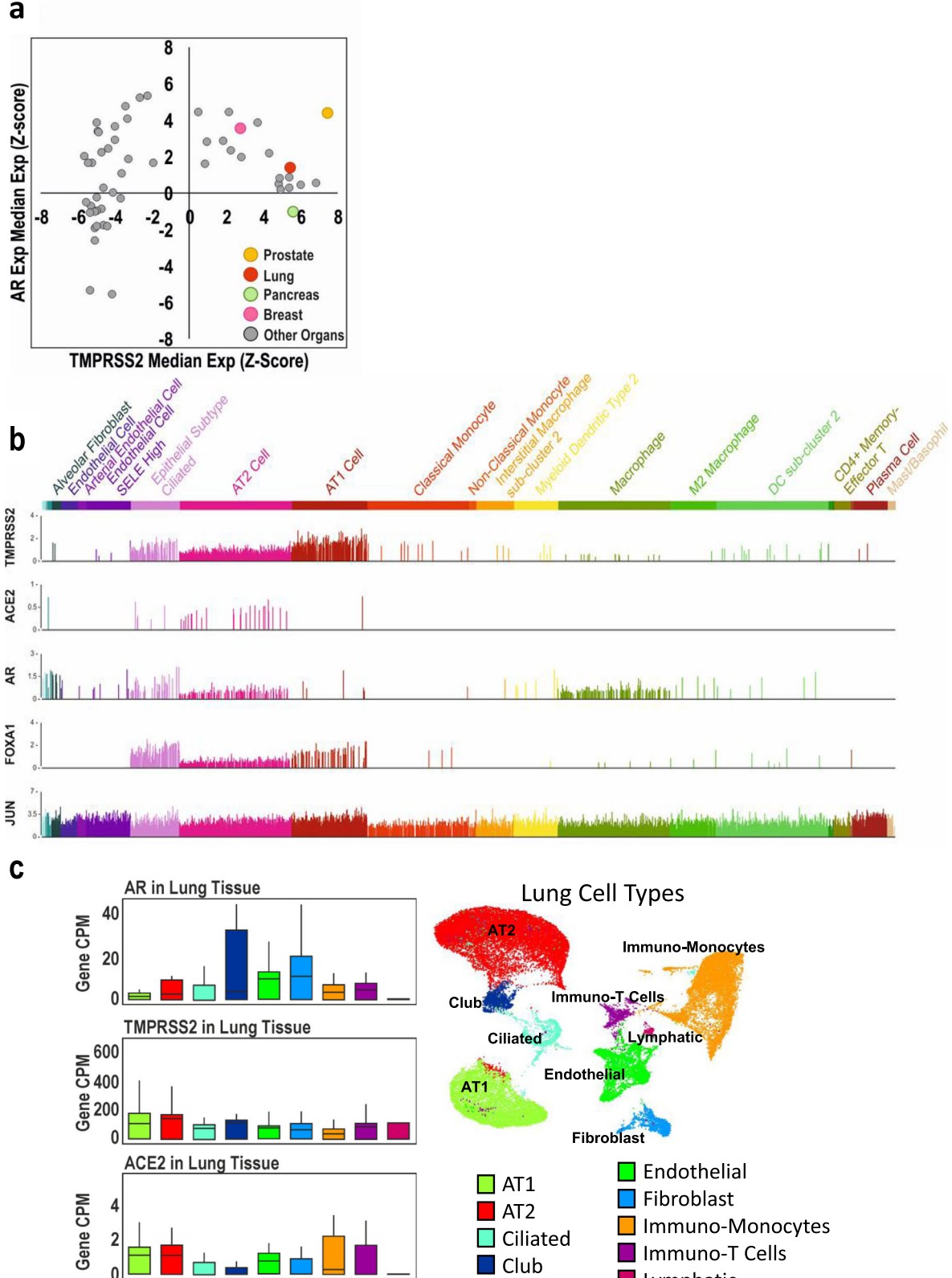

**Fig. 2 TMPRSS2 and the AR are expressed in the lung. a** The GTEx dataset[40] was interrogated and median expression of AR was plotted against median TMPRSS2 expression. Data are expressed as each gene normalised across all tissues (z-score). Highlighted are prostate (n = 245), lung (n = 578), pancreas (n = 328) and breast (n = 459). **b** Single-cell analysis of lung tissue[41] with each bar representing AR, TMPRSS2, ACE2, FOXA1, JUN expression within individual cells of each specified cell type. **c** Single-cell analysis of lung tissue[42] was interrogated and a breakdown is shown of AR, TMPRSS2 and ACE2 expression levels in specific lung cell types (X-axis); Y-axis denotes gene mRNA CPM.

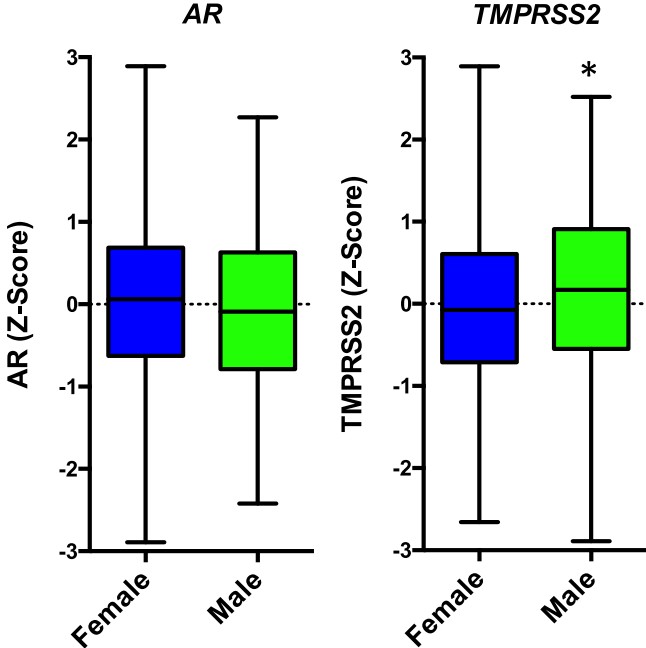

**AR** **TMPRSS2**

**Fig. 3 TMPRSS2 is expressed at higher levels in the male lung compared to female lung.** The GTEx dataset[40] was interrogated and expression of AR and TMPRSS2 investigated in lung tissue and dichotomised by gender, M = male (N = 220), F = female (N = 355). Box and whiskers: the centre line represents the median, the box represents the 25th and 75th percentile and the bars represent the min and max values. One-tailed t-test, *p = 0.0403.

and mice that had been castrated (removal of testicular production of androgen) were interrogated (GSE31341)[50]. In agreement with our cell line data, castration significantly reduced *Tmprss2* expression in the mouse lung (Fig. 5c). In the same mice, castration was also associated with an increase in *Ar* expression (p = 0.0018), expected as *Ar* gene transcription is downregulated in response to androgen[51]. In contrast to the results presented here, Baratchian et al. found no regulation of Tmprss2 in the mouse lung in response to ENZA[47]. However, in that study mice were fed ENZA whereas in this study oral gavage was used. The method of ENZA administration may therefore go some way to explain this discrepancy.

**Regulation of TMPRSS2 expression is tissue specific**. Although ChIP-Seq data for genomic AR binding in lung tissue or cells are not available, we were able to assess the cistrome of FOXA1 and JUN, known pioneer coregulatory factors for the AR[52] and other nuclear receptors. Binding of the glucocorticoid receptor (GR) was also investigated as this can bind to many of the same response elements as the AR[53], also acetylated histone 27 (H3K27ac) as an indicator of active regulatory regions, all in A549 lung cells (Fig. 6a). Binding profiles for prostate (LNCaP) and breast (MCF-7) cell lines were included for comparison. In LNCaP cells the AR binding pattern correlates with previous findings[24], and confirms that AR and GR bind in the same regions, corresponding also to binding of the pioneer factor FOXA1, and these sites largely correlate with the marker of transcriptionally active regions, H3K27ac. Detailed analysis of these potential response elements by the Claessens laboratory demonstrated that an androgen response element in the enhancer region (approximately –13 kb from the transcription start site) is crucial for optimal androgen regulation of *TMPRSS2* in prostate cells[24].

The binding patterns of GR, pioneer factors and H3K27ac in lung cells, however, differ to what is seen in LNCaP cells (compare regulatory region 1 and 2). To assess if androgen response elements are present in regulatory region 2, the AR binding motif (MA0007.2, Fig. 6b) from the JASPAR database was used to detect AR target sites using methods previously described[54]. This analysis identified potential androgen response elements throughout the 5' region of the *TMPRSS2* gene (Fig. 6a and Supplementary Table 1). Importantly, several of the potential androgen response elements were found to correlate with the GR, FOXA1, JUN, and H3K27ac peaks seen in the A549 regulatory region 2. Together, this suggests that AR (and associated factors) may directly regulate *TMPRSS2* via different regulatory regions in lung and prostate. To investigate this, we performed ChIP-qPCR on these regions in LNCaP, A549 and H1944 cells (Fig. 6c). In agreement with our prediction, the AR binding sites differed between lines, with AR predominantly binding to regulatory region 1 in LNCaP and regulatory region 2 in A549 and H1944. Li et al. also found AR binding upstream of TMPRSS2 to differ between prostate and lung lines[55]. They demonstrated AR binding to regulatory region 1 in LNCaP cells, whereas the AR did not bind to this region in the lung cells investigated, compatible with our analysis. However, that analysis did not expand as far upstream as regulatory region 2, so did not investigate whether the AR was present at regulatory region 2, as demonstrated here. The data therefore suggest that the AR may regulate TMPRSS2 expression differently in different tissues. Further analysis will be required to confirm that AR binding to these response elements regulates TMPRSS2 expression.

The DNA binding of AR, GR, FOXA1, JUN and H3K27ac around the TMPRSS2 gene in breast cancer cells (MCF-7) appears to be less pronounced than in prostate and lung, and the binding pattern has elements of the binding patterns in both prostate and lung cells. Importantly, AR binding in MCF-7 cells correlates with the H3K27ac, FOXA1, JUN and GR peaks located distally in the A549 regulatory region 2. Intriguingly, this region also correlates with a peak for oestrogen receptor-α (ESR1) binding in MCF-7, and oestrogen has been shown to downregulate TMPRSS2 expression[56]. This supports the possibility of TMPRSS2 regulation by other members of the nuclear receptor superfamily, and hence further potential for pharmacological manipulation—in this case by oestrogens as well as, via the GR, glucocorticoids.

**Antiandrogens can successfully reduce SARS-CoV-2 infection**. We have demonstrated that antiandrogens can successfully reduce TMPRSS2 expression in lung cells. To test our hypothesis that this will inhibit SARS-CoV-2 viral entry, A549 cells were treated with BIC or ENZA for 72 h prior to transduction for 48 h with SARS-CoV-2 Spike protein pseudotyped and luciferase-expressing lentivirus (Fig. 7a). Both antiandrogens significantly reduced viral entry, with the latter reducing cell entry by approximately 50%. In contrast, cellular entry driven by the glycoprotein of the pantropic vesicular stomatitis virus G protein (VSV-G) was unaffected by the antiandrogens (Fig. 7a), demonstrating that the effects of these drugs are specific for the SARS-CoV-2 mechanism of cell entry. To confirm that ENZA downregulates TMPRSS2 expression under these conditions (cells grown in full media) at the point of viral transduction, the SARS-CoV-2 Spike pseudotyped experiment was repeated ± ENZA (Supplementary Fig. 3a–c) and qPCR and immunoblotting performed to assess TMPRSS2 levels. As expected, TMPRSS2 expression was downregulated in response to the antiandrogen at the RNA and protein level. To see if antiandrogens could inhibit SARS-CoV-2 infection, $TCID_{50}$ assays were performed with the virus (Fig. 7b). For this, A549 cells were transfected with a vector expressing ACE2 to facilitate the infection[57], and treated with the

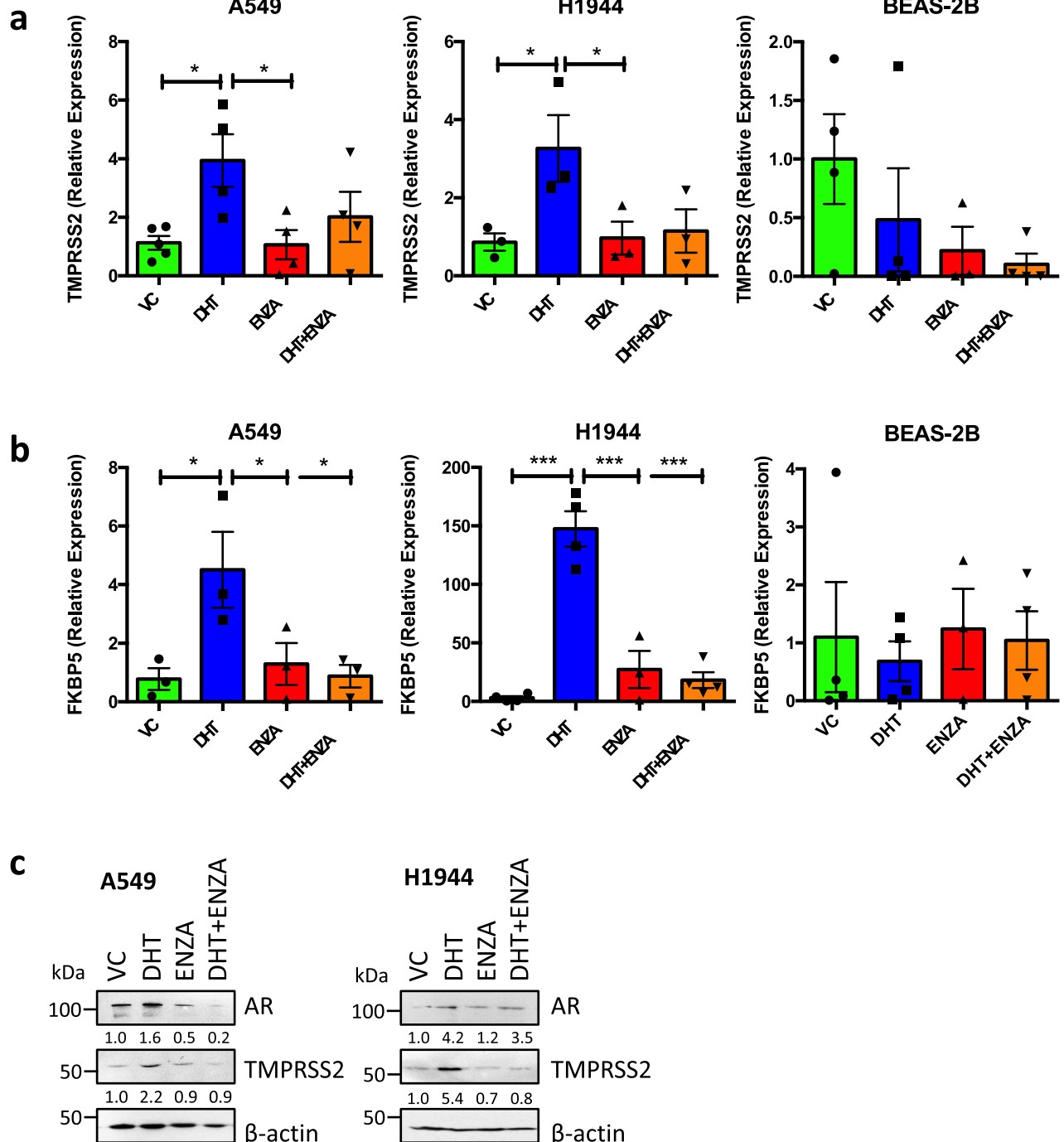

**Fig. 4 TMPRSS2 is androgen regulated in lung cell lines.** A549, H1944 and BEAS-2B were incubated in hormone-depleted media for 72 h and treated with ±10 nM dihydrotestosterone (DHT) ± 10 μM enzalutamide (ENZA) for 24 h. RNA was harvested, reverse transcribed and qPCR performed to quantify **a** *TMPRSS2* and **b** *FKBP5* expression. Mean of at least three independent repeats (±1 SEM). Significance determined using one-way ANOVA with Dunnett's multiple comparison test. TMPRSS2: A549 Vehicle Control (VC) v DHT $p = 0.0190$, DHT v ENZA $p = 0.0228$; H1944 VC v DHT $p = 0.0410$, DHT v ENZA $p = 0.0497$. FKBP5: A549 VC v DHT $p = 0.0250$, DHT v ENZA $p = 0.0493$, DHT v DHT + ENZA $p = 0.0285$; H1944 VC v DHT $p = 5.7 \times 10^{-6}$, DHT v ENZA $p = 7.1 \times 10^{-5}$, DHT v DHT + ENZA $p = 1.7 \times 10^{-5}$. **c** A549 were incubated in hormone-depleted media for 72 h and treated with ±10 nM DHT ± 10 μM ENZA for 24 h. Cells were lysed and proteins separated using SDS-PAGE. Immunoblotting was performed to visualise AR and TMPRSS2 expression levels and β−actin used as a loading control. Densitometry was performed for AR and TMPRSS2, values normalised to β−actin and made relative to VC. Representative immunoblots blots – A549 $n = 3$, H1944 $n = 2$. Source data are provided as a Source Data file.

antiandrogens for 72 h prior to infection. In agreement with the pseudotyped virus experiments, the antiandrogens significantly reduced SARS-CoV-2 viral titres by approximately 18-fold for ENZA and 13-fold for BIC.

The data demonstrate that antiandrogens inhibit viral entry and infection, and this is likely, at least in part, to be as a result of downregulation of TMPRSS2. However, antiandrogens may also regulate the expression of other factors important in viral entry.

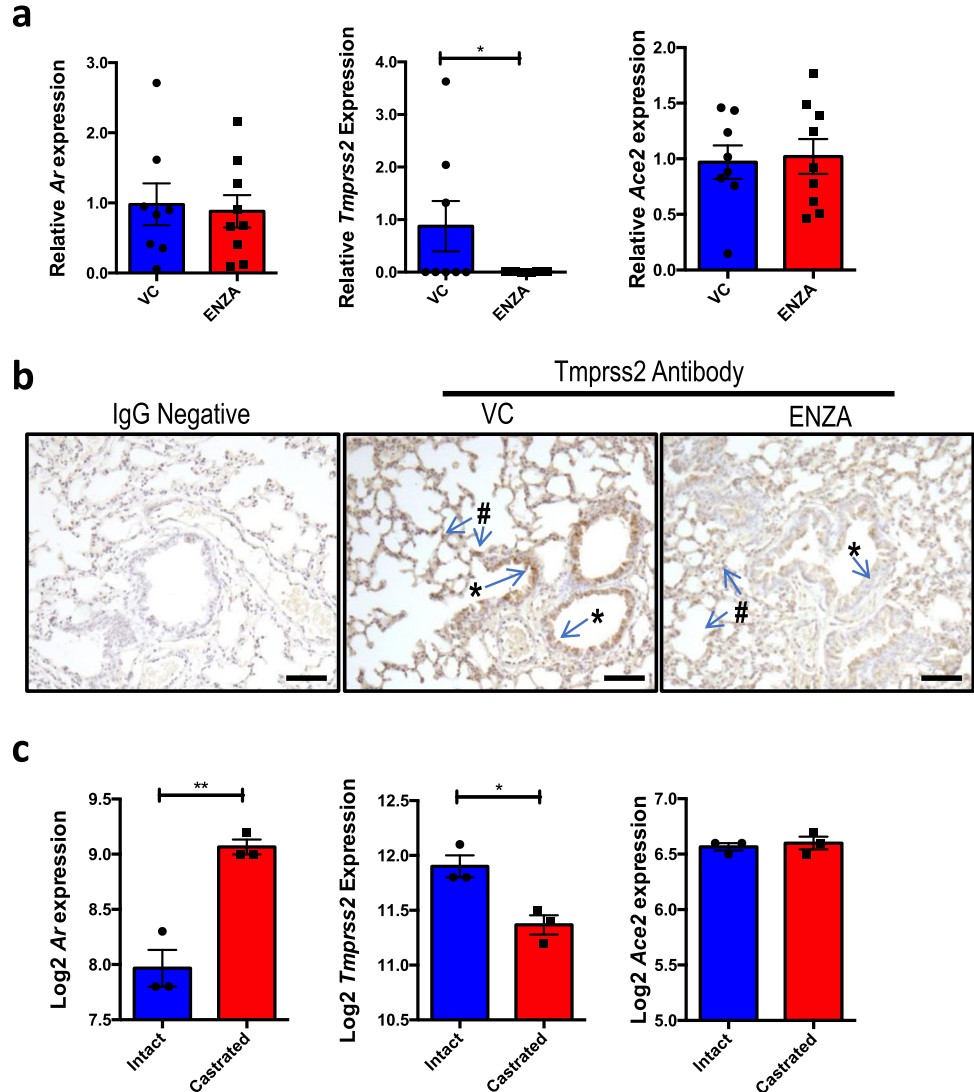

**Fig. 5 Enzalutamide downregulates Tmprss2 in mouse lung. a** Relative expression of *Ar*, *Tmprss2* and *Ace2* mRNA from lung tissue of mice treated with enzalutamide (ENZA, *n* = 9) or vehicle control (VC, *n* = 8) once daily for 3 days. Expression was normalised to housekeeping genes and made relative to VC. Mean ± 1 SE. Student's *t*-test (one-tailed) \**p* = 0.0367. **b** Example of Tmprss2 IHC in mouse lung. Representative immunostaining of epithelial cells (\*) and parenchyma/AT cells (#) in sections from *n* = 9 VC-treated mice, and *n* = 8 ENZA-treated mice. Bar = 50 μm **c** Log2 expression of *Tmprss2*, *Ar* and *Ace2* in lung tissue from male mice physically castrated (*N* = 3) vs intact male mice (*N* = 3) (GSE31341)[50]. Mean ± 1 SE. Student's *t*-test (one-tailed), \*\**p* = 0.0018, \**p* = 0.0081. Source data are provided as a Source Data file.

For example, it was recently demonstrated that antiandrogens can also downregulate expression of ACE2[44]. It is therefore possible that the inhibitory effects of antiandrogens may be as a result of dual downregulation of TMPRSS2 and ACE2. However, antiandrogens were effective in the SARS-CoV-2 experiments, in cells with exogenous ACE2 over-expressed from a promoter without the endogenous regulatory regions so likely not subject to antiandrogen regulation, supporting that the downregulation of TMPRSS2 plays a pivotal role in the inhibitory effects of these antagonists.

The incomplete block of viral entry/infection could be a result of S-protein priming via different proteases or due to alternative, currently unknown, virus entry mechanisms. For example, TMPRSS4 (not a known AR target gene) has also been shown to facilitate SARS-CoV-2 cell entry in the lung[58] and therefore targeting additional proteases involved in S-priming may have additive/synergistic effects.

Since the initial submission of this manuscript, several studies investigating AR as a potential target for COVID-19 therapy have been published. Contrary to the findings here, Li et al. did not find TMPRSS2 to be androgen-regulated in lung cell lines[55]. This may be due to the cell lines used in their study (H2126 and H1437) differing to the lines used here. The group also found that ENZA was unable to reduce SARS-CoV-2 infectivity in human lung organoid models. Characterisation of the different cell types present, and confirmation that the AR is functional in these organoids, would assist in explaining the discrepancy between these findings and our own. However, several other studies do support the use of androgen signalling inhibitors as treatment options for COVID-19. For example, Qiao et al. demonstrated that androgens regulate TMPRSS2 expression in subsets of lung epithelial cells[44], while Deng et al. demonstrated that TMPRSS2 is androgen-regulated in H460 lung cells, and that ENZA successfully reduces TMPRSS2 expression and blocks viral entry[59]. Furthermore, Samuel et al. found that 5α-reductase inhibitors (inhibitors of DHT synthesis) reduced TMPRSS2 (and ACE2) expression and viral infection in lung organoids[60].

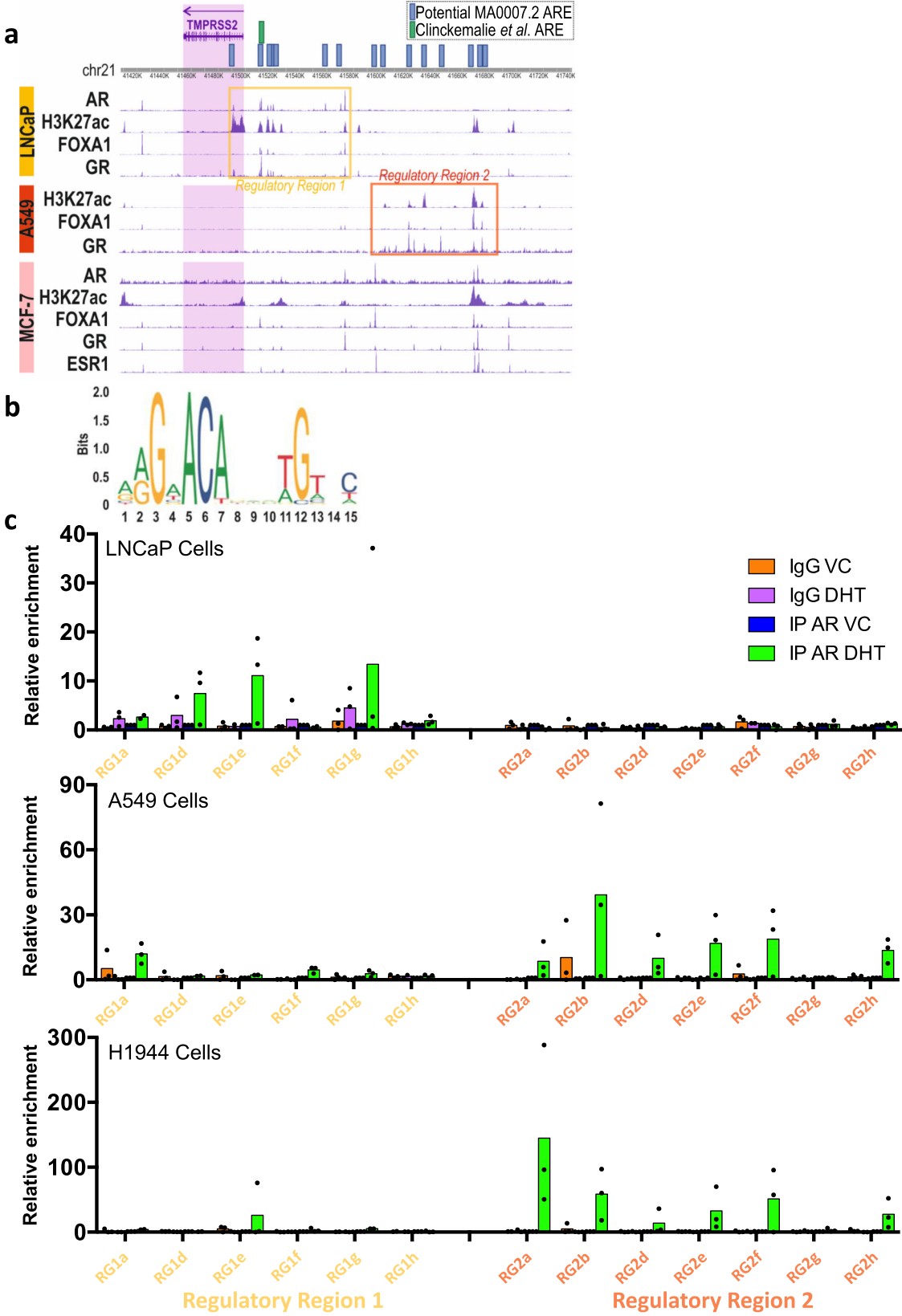

The data presented and literature discussed here suggest a role for AR in regulation of TMPRSS2 in the lung, which may at least in part explain why men with COVID-19 have a worse prognosis compared to women. Data from prostate and breast tissue also support regulation in other organs, which may also be targeted by SARS-CoV-2. Importantly, our findings support the hypothesis that therapies to target AR signalling could be used to transcriptionally inhibit TMPRSS2 expression, in lung and other organs. Furthermore, potential regulation of TMPRSS2 by other, related receptors (revealed by cistromic analysis) opens up the

**Fig. 6 Potential regulatory regions in the TMPRSS2 promoter. a** ChIP-sequencing peaks of AR, FOXA1, GR and H3K27ac in LNCaP cells; FOXA1, GR and H3K27ac in A549 lung cells; AR, H3K27ac, FOXA1, GR, ESR1 in MCF-7 breast cancer cells. The *TMPRSS2* gene is highlighted in the purple shaded box and the potential regulatory region 1 is boxed in yellow, the potential regulatory region 2 is boxed in orange. Potential AREs are marked in blue boxes (MA0007.2) or green (determined by[24]). Position of potential MA0007.2 motifs around the TMPRSS2 gene separated into two regions, the first region (regulatory region 1) covering areas of TSS and promoter/enhancers, the second region (regulatory region 2) covers more distant enhancer regions. **b** AR motif MA0007.2 from JASPAR curated motif database. **c** ChIP-qPCR of AR binding, and IgG control, to AR-motif-containing sites within regulatory regions 1 and 2 in LNCaP, A549, and H1944 cells treated with 10 nM dihydrotestosterone (DHT) or vehicle control (VC) for 4 h. Data are normalised to input and made relative to AR-IP VC. Data are the mean of three independent repeats. A maximum of one outlier was removed from each treatment using Grubbs' test (alpha = 0.05). Source data are provided as a Source Data file.

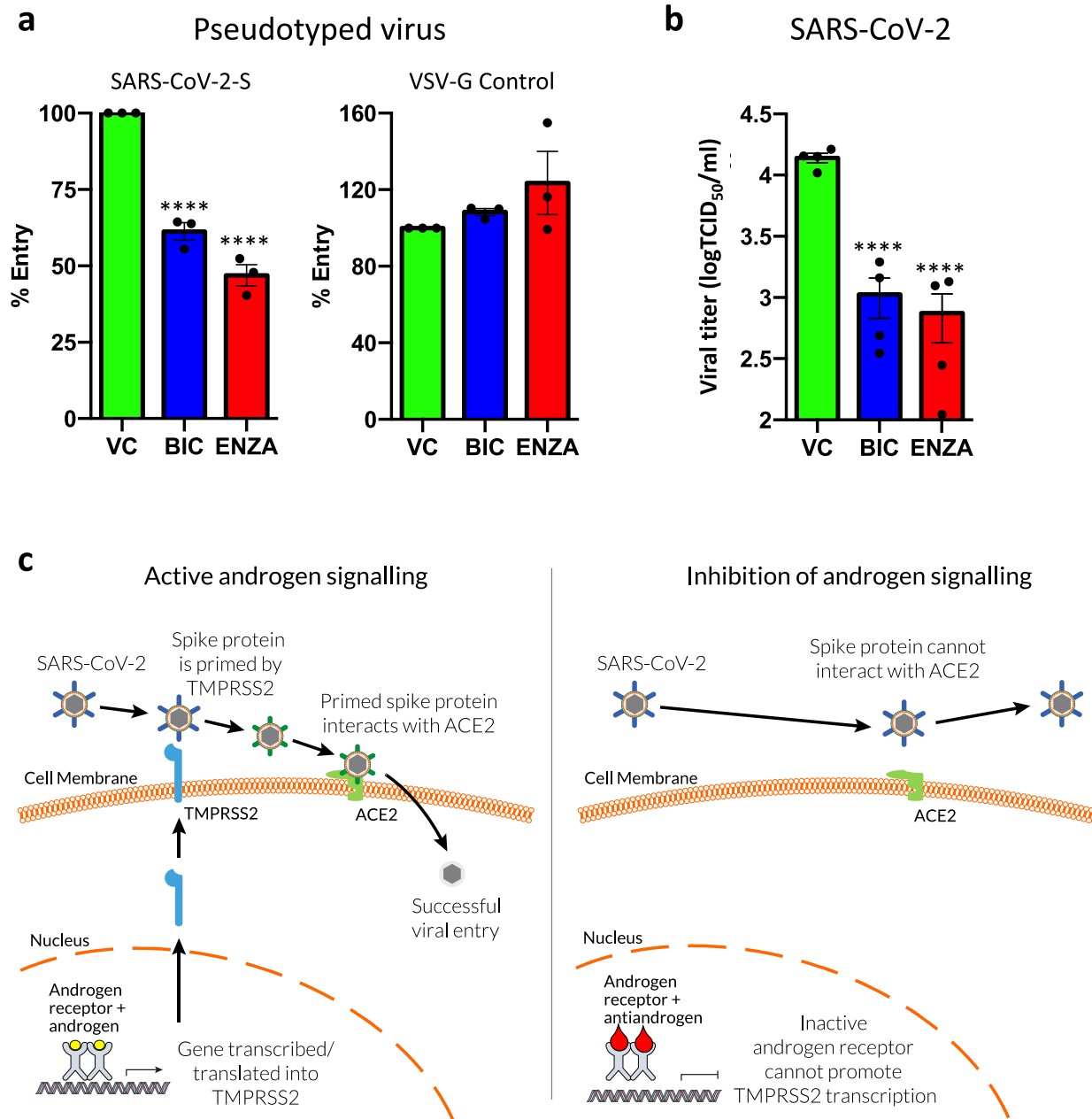

**Fig. 7 Antiandrogens reduce SARS-CoV-2 entry and infection in lung cells. a** A549 cells were seeded in full media and treated with ±10 μM bicalutamide (BIC) or enzalutamide (ENZA) for 72 h prior to transduction with SARS-CoV-2 Spike protein or VSV-G control pseudotyped lentiviral particles expressing luciferase. Cells were left for an additional 48 h and luciferase assays performed. Luciferase data were normalised to total protein content. Mean of three independent repeats ± 1 SE. One-way ANOVA with Sidak's multiple comparison test. Vehicle Control (VC) v BIC $p = 1.3 \times 10^{-4}$, VC v ENZA $p = 2.1 \times 10^{-5}$. **b** A549 were seeded in full media, transfected with ACE2 and treated with ±10 μM BIC or ENZA for 72 h prior to infection with SARS-CoV-2. Infectious titres of SARS-CoV-2 in supernatants of A549-ACE2 were determined with TCID$_{50}$ assays after 24 h of infection. $n = 4$, mean of two independent experiments repeated in duplicate ± 1 SE. One-way ANOVA with Sidak's multiple comparison test. VC v BIC $p = 2.0 \times 10^{-6}$, VC v ENZA $p = 2.5 \times 10^{-6}$. **c** Schematic representation of how targeting the AR could reduce SARS-CoV-2 entry. Source data are provided as a Source Data file.

possibility of additional potential opportunities for pharmacological inhibition of TMPRSS2 expression. Downregulation of TMPRSS2 will result in attenuated spike protein priming, reducing SARS-CoV-2 interaction with ACE2 and viral entry (summarised in Fig. 7c). Antiandrogens are used routinely in, or have been trialled for, the treatment of multiple diseases, including prostate cancer, breast cancer, polycystic ovarian syndrome and alopecia[61]. They have been shown to be well tolerated in men and women[61–63] and therefore antiandrogen treatments should be considered as potential therapeutic, and possibly preventative, strategies for COVID-19.

## Methods

**Ligands.** Mibolerone was purchased from Perkin Elmer (Waltham, MA, USA) and dissolved in ethanol. DHT/5alpha-Androstan-17beta-ol-3-one was purchased from Sigma Aldrich (A8380) and dissolved in ethanol. ENZA was from Sigma Aldrich (St. Louis, MO, USA) and was dissolved in DMSO.

**Cell culture.** A549, GMK (Vero E6), 293T, LNCaP, H1944 and BEAS-2B were purchased from the ATCC (Manassas, VA, USA). A549 and 293T cells were cultured in Dulbecco's Modified Eagle's Medium (DMEM, Sigma Aldrich), LNCaP and H1944 in Roswell Park Memorial Institute (RPMI, Sigma Aldrich) medium 1640 supplemented with 10% FCS and 2 mM L-glutamine, 100 U/ml penicillin, 100 mg/ml streptomycin (PSG)[64]. BEAS-2B were grown in Bronchial Epithelial Cell Growth Medium (Lonza, Basel, Switzerland). GMK were grown in DMEM supplemented with 10% FCS, 1% non-essential amino acids (NEAA) and PS. For experiments involving hormone manipulation, cells were incubated in phenol red free DMEM or RPMI (Invitrogen, Carlsbad, CA, USA), supplemented with double charcoal-stripped FCS and PSG (First Link UK Ltd., Wolverhampton, UK).

**Immunoblotting.** Cells were lysed in RIPA buffer (150 mM NaCl, 1% IGEPAL CA-630, 0.5% sodium deoxycholate, 0.1% SDS, 25 mM Tris pH 7.4) containing freshly added protease inhibitors (Halt Protease Cocktail, Thermo Fisher, Waltham, MA, USA). Samples were incubated on ice for 10 min, sonicated 4 cycles of 30 s on/off (Bioruptor, Diagenode, Denville, NJ, USA) and centrifuged (21,100 ×g, 10 min, 4 °C). The DC protein assay (BioRad, Hercules, CA, USA) was used to quantify protein concentrations. Then, 30 μg of protein was separated using SDS-PAGE and immunoblotting performed[65]. Primary antibodies used were rabbit anti-TMPRSS2 (1:100, ab92323, Abcam, Cambridge, UK), mouse anti β-actin (1:8000, ab8226, Abcam), rabbit anti-AR (1:2000, ab74272, Abcam) and mouse α-tubulin (1:10,000, B-5-1-2, Sigma Aldrich). Secondary HRP-conjugated antibodies (Sigma Aldrich) were used at 1:2000. Proteins were visualised using Immobilon Forte HRP substrate (Merck Millipore, MA, USA) and a Fusion FX Chemi Imager (Vilber Lourmat, Collégien, France).

**Quantitative PCR analysis of gene expression in cell lines.** Cells were seeded in 12-well plates and incubated with +10 nM DHT ± 10 μM BIC or ENZA for 6 or 24 h. RNA was extracted using Trizol (Thermo Fisher), following the manufacturer's instructions. Then, 250 ng of RNA was reverse transcribed using a LunaScript RT SuperMix Kit (NEB, Ipswich, MA, USA). Alterations in gene expression were quantified using the Luna Universal qPCR Master Mix (NEB) and a Roche 96 qPCR machine (Basel, Switzerland). *TMPRSS2* data were normalised to *L19* data and the $2^{(-\text{delta delta CT})}$ method was used to calculate gene expression changes. Primer sequences are provided in Supplementary Table 2.

**Mouse studies.** $Pten^{\text{loxp/loxp}};Pb\text{-}Cre4$ mice (The Jackson Laboratory, Bar Harbour, ME, USA), which have prostate-specific PTEN deletion[66], were treated with 50 mg/kg ENZA (in 5% DMSO + 1% CMC + 0.1% P80 oral gavage) or vehicle control every day for 3 days. All work was carried out in accordance with the provisions of the Animals (Scientific Procedures) Act 1986 of the United Kingdom. The work was approved by Imperial College Animal Welfare and Ethical Review Body and performed under an appropriate Home Office license (PPL70_8705). Organs were snap frozen until RNA was extracted using a Monarch RNA extraction kit (NEB) following disruption of frozen tissue utilising mechanical disruption and enzymatic digestion with Proteinase K. Then, 2 μg of RNA was reverse transcribed using Precision Nanoscript2 RT Kit (Primer Design, Southampton, UK). Changes in gene expression were measured using SYBR Green Fast Master Mix (Life Technologies, Carlsbad, CA, USA) and QuantStudio 7 Flex Real-Time PCR System (Applied Biosystems, Foster City, CA, USA). Gene expression data were normalised to *L19*, *Gapdh* and *Actb* data and the $2^{(-\text{delta delta CT})}$ method used to calculate changes in expression.

**TMPRSS2 expression in tissue and cell line datasets.** RNA-sequencing dataset v8 was downloaded from the Genotype Tissue Expression (GTEx) Project Online Portal[67]. Gene expression was normalised (inverse normal transformation) across

samples, and medians for AR and TMPRSS2 expression across each tissue were calculated. Data from RNA sequencing of isolated single nuclei, performed on surgical specimens of healthy, non-affected lung tissue from 12 lung adenocarcinoma patients, were analysed for AR, TMPRSS2 and ACE2 expression using Eils Lab UCSC Cell browser (https://eils-lung.cells.ucsc.edu)[42]. Sequencing data from T47D cells treated with 10 nM DHT (GSE62243)[39], and data from lungs of castrated or intact mice (GSE31341)[50] were log2 transformed. Two single-cell seq datasets were investigated. The first dataset of lungs from 9 patients (GSE122960[41]) was analysed using http://altanalyze.org/, and the second dataset of lungs from 12 donors[42] was analysed using https://eils-lung.cells.ucsc.edu.

**Identification of potential AREs.** The AR binding motif, MA0007.2 was obtained from the JASPAR online curated motif database[54]. Segments of DNA around the TMPRSS2 gene were scanned for potential matches for presence of the MA0007.2 AR motif using FIMO[68]. Of the 40 possible MA0007.2 matches in regulatory region 1 ($p < 0.0001$) and 34 possible matches in regulatory region 2 ($p < 0.0001$), only response elements that correlated with binding peaks were selected.

**Analysis of ChIP-sequencing data.** ChIP-sequencing data (receptors in the presence of their cognate ligands) were analysed using Cistrome Analysis Pipeline and visualised with WashU Epigenome Browser (v51.0.5). Datasets used: LNCaP – AR (GSE94682)[69], H3K27ac (GSE73783)[70], FOXA1 (GSE94682)[69] and NR3C1/GR (GSE39880)[71]; A549 – H3K27ac (GSE29611)[72], FOXA1 (GSE32465)[73] and NR3C1/GR (GSE91313)[72]; MC7 – AR (GSE104399)[74], H3K27ac (GSE94804)[75], FOXA1 (GSE112969)[76] and NR3C1/GR (GSE39879)[71].

**ChIP-qPCR.** LNCaP, A549 and H1944 cells were grown in 15 cm plates to approximately 80% confluency, before 4 h treatment with 10 nM DHT or vehicle control. Cells were fixed with 4% formalin and ChIP performed. Antibodies used in overnight 4 °C immunoprecipitation were anti-AR (sc-7305, Santa Cruz Biotechnology, Santa Cruz, CA, USA) and rabbit mouse IgG at 10 μg per sample. After DNA isolation with phenol:chloroform:isoamyl alcohol and resuspension in water, enrichment across the two potential TMPRSS2 regulatory regions was quantified with qPCR (primers in Supplementary Table 2).

**Immunohistochemistry.** Lung samples from the above mouse studies were fixed in 4% formalin for 24 h, before transfer to 70% alcohol for storage before processing into wax. Sections were probed with anti-Tmprss2 antibody (ab92323, Abcam) and anti-AR (ab108341, Abcam) overnight at 4 °C at a 1:1000 dilution before detection with the Histostain-Plus IHC HRP Kit and DAB (Thermo Fisher).

**Quantification of cell entry using pseudotyped virus.** SARS-CoV-2 Spike protein and VSV-G pseudotyped lentiviral particles expressing luciferase were produced in 293T cells cultured in a 6-well plate. For this, 1 μg pCAGGS (GAG/POL), 1.5 μg pCSFLW (Luciferase) and 1 μg pcDNA3.1-SARS2-Spike or pcDNA3.1-VSV-G were transfected into 293T cells using FugeneHD (Promega, Madison, WI, USA) at a ratio of 3 (lipid):1 (DNA) following the manufacturer's instructions. The lentiviral packaging constructs pCSFLW and pCAGGs-GAGPOL have been previously described[78] and the codon-optimised pcDNA3.1-SARS2-Spike and pcDNA3.1-VSV were kind gifts from Professor Robin Shattock (Imperial College London)[79] and Dr Nigel Temperton (University of Kent)[80], respectively. After 5 h, medium was changed and after another 72 h, the medium supernatant was harvested and centrifuged. The titre of lentiviral particles was measured with a HIV-1 Gag p24 ELISA (Bio-Techne, Minneapolis, MI, USA) according to the manufacturer's protocol. A549 cells were grown in 48-well plates in full media with ±10 μM BIC or ENZA for 72 h prior to transduction with $3.75 \times 10^4$ TU per well. Cells were left for an additional 48 h and viral entry quantified using luciferase assay (E4030 Luciferase Assay System, Promega) according to the manufacturer's instructions. Luciferase activity was normalised to total protein content, quantified using a NanoDrop UV Spectrophotometer (Thermo Fisher).

**SARS-CoV-2 infection studies.** A549 cells were seeded in 12-well plates and transiently transfected with 1 μg of pCAGGS-ACE2 (synthesised by GeneArt, Thermo Fisher) using Lipofectamine 3000 reagent as described by the manufacturer (Thermo Fisher). After 24 h, cells were treated with ±10 μM BIC or ENZA for 56–72 h. Cells were then washed with PBS and infected with SARS-CoV-2. The viral strain was SARS-CoV-2/England/IC19/202 (IC19)[79]. All work involving the use of SARS-CoV-2 was performed in a Containment Level 3 (CL3) laboratory (St Mary's, Imperial College London).

The virus was diluted in serum-free DMEM (supplemented with 1% NEAA and PS) to a multiplicity of infection of 1. The inoculum was added to A549 cells overexpressing ACE2 and incubated at 37 °C for 1 h. The inoculum was then removed and cells maintained as described above. Twenty-four-hour post infection, the culture supernatants were collected and quantified by TCID50 assay on GMK Vero E6 cells[11]. Serial dilutions of the virus (in serum-free DMEM) were added in 96-well plates and cells were left for 4–5 days before they were fixed with 2× crystal violet solution and analysed. TCID50 titres were determined by the Spearman–Karber method[81].

**Statistical analyses.** Statistical analyses were performed using GraphPad PRISM (v 6.0c). For experiments with two treatment arms, one-tailed $t$-tests were performed. For analysis of more than two treatments, one-way or two-way ANOVA was performed with Sidak's, Dunnett's or Tukey's multiple comparison tests. For comparison of AR and TMPRSS2 levels in the human lung (GTEx dataset), Wilcoxon tests were performed. All experiments were at least three independent repeats, unless otherwise stated.

**Reporting summary.** Further information on research design is available in the Nature Research Reporting Summary linked to this article.

## Data availability

For analysis of TMPRSS2 in different tissues, the GTEx dataset was used (https://www.gtexportal.org/home/). Transcriptomic analysis of gene expression was performed on datasets obtained from GEO: T47D cells treated with 10 nM DHT (GSE62243); data from lungs of castrated or intact mice (GSE31341). ChIP-Seq data were obtained from GEO: LNCaP – AR (GSE94682), H3K27ac (GSE73783), FOXA1 (GSE94682) and NR3C1/GR (GSE39880); A549 – H3K27ac (GSE29611), FOXA1 (GSE32465) and NR3C1/GR (GSE91313); MC7 – AR (GSE104399), H3K27ac (GSE94804), FOXA1 (GSE112969) and NR3C1/GR (GSE39879). sc-Seq – AR, TMPRSS2 and ACE2 expression were analysed using Eils Lab UCSC Cell browser (https://eils-lung.cells.ucsc.edu). Two additional single-cell datasets were investigated (GSE122960). The remaining data are available within the article and the Supplementary information. Source Data are provided with this paper.

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

## Acknowledgements

D.A.L. was funded by the Imperial College London COVID Fund and Prostate Cancer Foundation and A.M.I. was funded by the University of Essex PhD Scholarship scheme. E.C. was funded by the Turkish Ministry of National Education. D.A.L., C.L.B., G.N.B., A.M. and R.M.Z. are funded by Prostate Cancer UK. We are grateful to Robin Shattock and Nigel Temperton for kind gifts of plasmids. We thank Piers Aitman for discussion and image design/processing and Gilberto Serrano de Almeida for assistance with in vivo experiments. We also thank the Brooke and Bevan labs and Antonio Marco for discussion of the project and Lynwen James for assistance with cell culture. Finally, we are grateful for infrastructure support provided by the CRUK Imperial College Centre and University of Essex. The GTEx project is supported by multiple funding bodies NIH, NCI, NHGRI, NHLBI, NIDA, NIMH and NINDS.

## Author contributions

D.A.L., A.M., E.S.G., E.C., A.M.I., L.L.Y., R.M.Z. and G.N.B. performed experiments. D.A.L, W.S.B., R.M.Z., C.L.B. and G.N.B. designed experiments and interpreted data. D.A.L., C.L.B. and G.N.B. conceived the study and co-wrote the manuscript.

## Competing interests

The authors declare no competing interests.
