## [Peer Review File · Nature Communications]

REVIEWER COMMENTS

Reviewer #1 (Remarks to the Author):

In this manuscript, Leach et al and colleagues studied androgen receptor (AR) regulation of TMPRSS2, a key protein required for SARS-Cov-2 entry into the host cells, in lung tissues, and suggested a potential targeted therapy to treat Covid-19 with the use of a potent and clinically approved antiandrogen drug, enzalutamide. The AR regulation of TMPRSS2 was well documented in prostate cancer research and the best evidence is the androgen-regulated overexpression of TMPRSS2-ERG fusion gene, which fuses the 5'-untranslated region of TMPRSS2 (contains multiple well-characterized AREs) and exons of ERG. Given the specific and strong regulation of AR on TMPRSS2, the recent discovery of the requirement of TMPRSS2 for SARS-CoV-2 infection reveals a possibility that targeting AR signaling may be a potential therapeutic strategy to treat the Covid-19 pandemic, particularly in men with the disease. Therefore, this study is timely and conceptually innovative. However, the critical question in this emerging area is whether TMPRSS2 expression in lung cells is strongly regulated by AR signaling. Unfortunately, the experiments conducted in this study were largely superficial and incomplete, and the authors failed to demonstrate clear evidence for AR regulation on TMPRSS2 in lung cells. Several comments were listed below:

1. In Fig.2, showing AR mRNA expression is not sufficient to demonstrate transcriptional active AR protein. IHC staining of nuclear AR is generally a marker for active AR signaling.
2. In Fig. 4, AR expression in A549 cells is very weak. Can androgen treatment (such as DHT or R1818) induce AR expression in this cell line (androgen can stabilize AR protein) or activate other AR regulated genes such as FKBP5? It is also not clear why androgen treatment cannot induce TMPRSS2 expression in this cell line.
3. In Fig. 5, IHC staining should be used to examine the response of AR signaling and protein expression of TMPRSS2 to enzalutamide in lung tissues.
4. In Fig. 6, FOXA1 is a required pioneer factor for AR binding at TMPRSS2 enhancers. Therefore, lacking FOXA1 binding at the well-characterized AREs on TMPRSS2 upstream enhancers may suggest that the TMPRSS2 expression is not regulated by AR. Interestingly, the results in Fig.6A may indicate a possible regulation of FOXA1/GR on TMPRSS2 expression in lung cells.
5. A thorough study to demonstrate AR binding and enhancer activation at the TMPRSS2 gene should be conducted in this cell line or other lung cell lines with significant expression of TMPRSS2 and AR.
6. The authors should also use other AR signaling-targeted inhibitors in the study to compare the efficacy with enzalutamide.

Reviewer #2 (Remarks to the Author):

SARS-CoV-2, the cause of COVID19, is a highly contagious virus that primarily affects the respiratory system. Studies have shown that SARS-CoV-2, like SARS-CoV, relies on the expression ACE2 and TMPRSS2 for viral entry into host cells. TMPRSS2 is a transcriptional target of the androgen receptor in prostate cancer cells and is responsive to androgen deprivation therapy. In the present manuscript, the authors first confirmed that TMPRSS2 is responsive to androgen and AR inhibitor enzalutamide in a prostate and a lung cell line, and castrated mice. They examined existing dataset to show AR and TMPRSS2 expression in various tissues and lung cell types. Finally, they identified a potential AR-target regulatory element (#2) in A549 that may be responsible for TMPRSS2 expression. While the topic of the study is of high significance to the COVID19 pandemic, the study is very limited in its scope and most of the findings are confirmatory as detailed below. A direct role of AR and androgen in regulating TMPRSS2 expression in a panel of lung cell lines and various lung tissue cell type was not established. Some of the results are in contrast with publically available studies and the controversy was not explained.

Major concerns:

1. Figure 1 is confirmatory, showing androgen regulation of TMPRSS2 in prostate and breast cancer cell lines.
2. Figure 2: TMPRSS2 expression in various tissues and lung cell types have been previously reported in multiple studies using similar bioinformatics analyses of existing RNA-seq or single-cell sequencing data.
3. Figure 3: there are a number of studies in public domain looked at TMPRSS2 expression between male and female and some of the studies reported a lack of difference. This discrepancy was not

explained.

4. AR level in A549 is very low when compared to LNCaP, raising concerns about its sensitivity to androgen stimulation. Figure 2C further showed that TMPRSS2 levels in various cell types of lung tissue are not different, despite their differences in AR expression. It is not convincing that androgen/AR play major roles in the regulation of TMPRSS2 in lung.

5. Figure 4 confirms TMPRSS2 as an AR target gene in LNCaP with incremental expansion to one lung cell line A549. AR expression, TMPRSS2 level, and androgen responsiveness should be examined in a wide range of lung cell types.

6. Figure 5 examined TMPRSS2 and AR expression in mice treated with Enz or castrated and showing decreased TMPRSS2 levels. However, tissue samples were not analyzed by scRNA-seq to see the responsible cell types. In addition, this is in contrast to a study led by Nima Sharifi who reported a lack of response. This discrepancy was not discussed.

7. Figure 6 suggested a potential AR-responsive element (at -140kb) in A549 and MCF-7 cells that is distinct to the one (-13.5kb) found in LNCaP cells. However, AR ChIP-seq was not done in A549 to directly show its binding at the 2nd regulatory element. CRISPR should be used to demonstrate its ability to regulate TMPRSS2 expression functioning as a distal enhancer. A panel of lung cell lines representing various cell types should be examined.

8. Figure 6: as the 2nd regulatory element is also present in MCF-7 cells and is also bound by ESR1, it would suggest that estrogen will be able to turn on TMPRSS2 expression as well. This is in direct contrast to the hypothesis that TMPRSS2 level is associated with high COVID19 death in men. Indeed, there were studies reporting that TMPRSS2 is also receptive to estrogen signaling (Stopsack, Mucci et al. 2020).

Reviewer #3 (Remarks to the Author):

SARS-CoV-2 uses ACE2 and TMPRSS2 for entry into cells. TMPRSS2 is a cellular serine protease that is known to be expressed in an androgen-regulated fashion. Leach and colleagues investigated whether the regulation by androgen can be exploited for COVID-19 therapy. They confirm that TMPRSS2 expression in LNCaP cells is androgen-regulated and provide evidence that the same is true for other cell lines. Further, they demonstrate coexpression of TMPRSS2 and androgen receptor (AR) in human lung tissue and show that TMPRSS2 expression in males is higher as compared to females. In addition, anti-androgen-mediated downregulation of TMPRSS2 expression in LNCaP and A549 cells as well as in murine lungs is shown and evidence for TMPRSS2 expression in lung being regulated by nuclear receptor proteins and coregulators is provided. The results reported are of interest. However, some points remain open.

Major

It is important to investigate whether the anti-androgen-mediated reduction in TMPRSS2 mRNA expression results in reduced SARS-CoV-2 entry. These analyses can be conducted fast with SARS-CoV-2 spike protein-pseudotyped vectors.

Figure 1 is not convincing. The cell lines should be treated using the same protocol, including the same androgen.

Minor

“Coronaviruses have structural (Spike, Nucleocapsid, Matrix and Envelope)”. “Matrix” should read “Membrane”

“TMPRSS2 primes the viral Spike (or S) protein by cleaving it at two sites; this facilitates fusion of the viral and host membranes [12-14].” This statement is incorrect – S1/S2 is cleaved by furin and S2` is cleaved by TMPRSS2 – and the references are outdated. Please consider citing PMID: 32376634 and PMID: 32362314.

“Cellular entry is subsequently facilitated by ACE2, “. In the view of this reviewer, it has not been demonstrated that SARS-CoV-2 spike is first cleaved and then binds to ACE2. For SARS-CoV-1, the opposite has been suggested.

We would like to thank the reviewers for taking time to carefully review our manuscript and for the constructive feedback provided. We are delighted that they support the significance of the study and have responded to all of their specific comments below. In responding to these helpful comments we have strengthened the manuscript by the inclusion of additional experiments, in collaboration where appropriate with expert colleagues (A. Mohr, E. S. Giotis, L. L. Yates, W. S. Barclay and R. M. Zwacka) whom we have thus added to the author list.

Reviewer #1 (Remarks to the Author):

1. In Fig.2, showing AR mRNA expression is not sufficient to demonstrate transcriptional active AR protein. IHC staining of nuclear AR is generally a marker for active AR signaling.

We have added additional citations that support that AR is nuclear and active in the lung, see page 4 “the AR is expressed in human and murine lung, and shown to be active as indicated by regulation of AR-target genes in mouse lung³¹, nuclear localisation of AR in (male) human lung^{31, 48}, and activation of an androgen-responsive reporter gene in mouse lung³²”. IHC images of the mouse lungs ± enzalutamide, with nuclear AR highlighted, have been added, see Supplemental Figure 2 and page 5 “The Ar was also found to be expressed in these cell types, with nuclear localisation indicating active Ar, and enzalutamide treatment resulted in a marked decrease in Ar levels in the lung (Supplemental Figure 2).”

2. In Fig. 4, AR expression in A549 cells is very weak. Can androgen treatment (such as DHT or R1818) induce AR expression in this cell line (androgen can stabilize AR protein) or activate other AR regulated genes such as FKBP5? It is also not clear why androgen treatment cannot induce TMPRSS2 expression in this cell line.

We have repeated the experiments with A549 (and other lung lines) and do see induction of TMPRSS2 in response to DHT (Figure 4a). As suggested, we have also confirmed androgen regulation of FKBP5 in these cells (Figure 4b). We have confirmed that DHT stabilises AR in A549, and enzalutamide prevents this ligand-mediated stabilisation (Figure 4c). Please see page 4.

3. In Fig. 5, IHC staining should be used to examine the response of AR signaling and protein expression of TMPRSS2 to enzalutamide in lung tissues.

IHC staining has been performed and representative examples of TMPRSS2 down-regulation in response to enzalutamide are provided (Figure 5a).

4. In Fig. 6, FOXA1 is a required pioneer factor for AR binding at TMPRSS2 enhancers. Therefore, lacking FOXA1 binding at the well-characterized AREs on TMPRSS2 upstream enhancers may suggest that the TMPRSS2 expression is not regulated by AR. Interestingly, the results in Fig.6A may indicate a possible regulation of FOXA1/GR on TMPRSS2 expression in lung cells.

The AR and GR share response elements and hence it is plausible that TMPRSS2 is also regulated by GR (and other steroid receptors). We have commented on this in the manuscript: see page 5 “.. glucocorticoid receptor (GR) was also investigated as this can bind to many of the same response elements as the AR...” and see page 6 “This supports the possibility of TMPRSS2 regulation by other members of the nuclear receptor superfamily, and hence further potential for pharmacological manipulation – in this case oestrogens as well as, via the GR, glucocorticoids.”

5. A thorough study to demonstrate AR binding and enhancer activation at the TMPRSS2 gene should be conducted in this cell line or other lung cell lines with significant expression of TMPRSS2 and AR.

We have now performed ChIP-qPCR in LNCaP (prostate), A549 (lung) and H1944 (lung) cells (Figure 6C). The data confirms AR binding at the identified sites and supports our hypothesis that the AR utilises different regulatory regions in lung compared to prostate.

6. The authors should also use other AR signaling-targeted inhibitors in the study to compare the efficacy with enzalutamide.

We have now also included bicalutamide, another commonly used antiandrogen in Figures 1a, 7a and Supplementary Figure 1a. We saw similar results for bicalutamide and enzalutamide.

Reviewer #2 (Remarks to the Author):

1. Figure 1 is confirmatory, showing androgen regulation of TMPRSS2 in prostate and breast cancer cell lines.

In this figure, we demonstrate that *TMPRSS2* is an androgen target gene in prostate and breast and that antiandrogens can inhibit gene expression. We feel that this is an important confirmatory experiment to include, as the starting point for all subsequent experiments. We are happy to move this to Supplemental if felt to be a better place for it.

2. Figure 2: *TMPRSS2* expression in various tissues and lung cell types have been previously reported in multiple studies using similar bioinformatics analyses of existing RNA-seq or single-cell sequencing data.

Our aim is to present novel experimental data plus some new analysis of existing datasets to show that targeting the AR could be a viable treatment option for COVID-19. We have added analysis of an additional single cell-sequencing-dataset to investigate AR and *TMPRSS2* expression in the different lung cell types. Please see Page 3: “Analysis of single cell sequencing data from lung tissue⁴¹, demonstrated that *TMPRSS2*, *ACE2*, AR and AR-associated pioneer factors (JUN and *FOXA1*) are co-expressed in Epithelial Subtype Ciliated and Alveolar Type 2 (AT2) cells (Figure 2B). Similarly in a second single cell data set⁴², AT2 cells were among the resident lung cells with the highest expression of *TMPRSS2*, *ACE2*, and AR (Figure 2C). These cell types are targeted by SARS-CoV-2⁴³.”

3. Figure 3: there are a number of studies in public domain looked at *TMPRSS2* expression between male and female and some of the studies reported a lack of difference. This discrepancy was not explained.

We have added text to summarise these discrepancies and added additional citations, see page 4:

“Since *TMPRSS2* is an androgen-regulated gene, it has been proposed that elevated levels of *TMPRSS2* in the lung, as a result of higher levels of androgen, might explain this gender disparity and it was therefore hypothesised that *TMPRSS2* expression would be higher in male lungs compared to females. Our analysis of the GTEx dataset found no significant difference in AR expression levels between men and women. *TMPRSS2* expression was also found to be similar between males and females (Figure 3), but expression in the male lung was found to be slightly and significantly higher, in agreement with a previous study⁴⁵. This is however, in contrast to other studies that have found no significant difference in *TMPRSS2* expression in male and female lungs^{37, 46, 47}. Further, Baratchian et al. found higher levels of *ACE2* in the male lung and proposed that alterations in the levels of this receptor could, at least in part, explain the gender disparity in COVID-19 severity⁴⁶. It therefore remains unclear if gender differences in *TMPRSS2* expression could explain why men suffer more severe symptoms following infection with SARS-CoV-2.

4. AR level in A549 is very low when compared to LNCaP, raising concerns about its sensitivity to androgen stimulation. Figure 2C further showed that *TMPRSS2* levels in various cell types of lung tissue are not different, despite their differences in AR expression. It is not convincing that androgen/AR play major roles in the regulation of *TMPRSS2* in lung.

Please see response to Reviewer 1, points 1-4.

5. Figure 4 confirms *TMPRSS2* as an AR target gene in LNCaP with incremental

expansion to one lung cell line A549. AR expression, TMPRSS2 level, and androgen responsiveness should be examined in a wide range of lung cell types.

We have expanded our analysis to include additional lung cell lines H1944 and BEAS-2B. Similar to A549, we see a significant increase in TMPRSS2 expression in response to DHT. In BEAS-2B, enzalutamide decreased TMPRSS2 expression, although this did not reach significance. Fig 5 also shows TMPRSS2 regulation *in vivo* in lung tissue.

6. Figure 5 examined TMPRSS2 and AR expression in mice treated with Enz or castrated and showing decreased TMPRSS2 levels. However, tissue samples were not analyzed by scRNA-seq to see the responsible cell types. In addition, this is in contrast to a study led by Nima Sharifi who reported a lack of response. This discrepancy was not discussed.

We have now performed IHC analysis of mouse lungs treated \pm ENZA and representative examples are provided (Figure 5b). Lung expert Dr Laura Yates (National Heart and Lung Institute) confirmed down-regulation of TMPRSS2 in response to ENZA and identified the cell types concerned, summarised on page 5 of the manuscript. It is true that the Sharifi preprint (Baratchian et. al, 2020) reports no response of TMPRSS2 to enzalutamide in mouse lung. However, as stated in the manuscript, published papers (e.g. Mikkonen et al., 2010) have previously reported TMPRSS2 upregulation by androgen in lung cells and downregulation in mouse lung following castration, supporting its regulation by AR, which would make it surprising were it not to be affected by antiandrogen treatment. We have also added the following sentence to acknowledge the disparity in results – please see page 5 “In contrast to the results presented here, Baratchian et al. found no regulation of Tmprss2 in the mouse lung in response to enzalutamide⁴⁶. However, in that study mice were fed enzalutamide whereas in this study oral gavage was used. The method of enzalutamide administration may therefore go some way to explain this discrepancy.”

7. Figure 6 suggested a potential AR-responsive element (at -140kb) in A549 and MCF-7 cells that is distinct to the one (-13.5kb) found in LNCaP cells. However, AR ChIP-seq was not done in A549 to directly show its binding at the 2nd regulatory element. CRISPR should be used to demonstrate its ability to regulate TMPRSS2 expression functioning as a distal enhancer. A panel of lung cell lines representing various cell types should be examined.

Please see response to Reviewer 1, point 5, ChIP-qPCR was performed and confirmed AR binding at these regions in different lung cell lines.

8. Figure 6: as the 2nd regulatory element is also present in MCF-7 cells and is also bound by ESR1, it would suggest that estrogen will be able to turn on TMPRSS2 expression as well. This is in direct contrast to the hypothesis that TMPRSS2 level is associated with high COVID19 death in men. Indeed, there were studies reporting that TMPRSS2 is also receptive to estrogen signaling (Stopsack, Mucci et al. 2020).

Wang et al. 2020 have shown that TMPRSS2 is down-regulated in response to oestrogen and this has been added to the text: see page 6 “Intriguingly, this region also correlates with a peak for oestrogen receptor- α (ESR1) binding in MCF-7, and oestrogen has been shown to down-regulate TMPRSS2 expression⁵⁴. This supports the possibility of TMPRSS2 regulation by other members of the nuclear receptor superfamily, and hence further potential for pharmacological manipulation – in this case oestrogens as well as, via the GR, glucocorticoids.” Please also see response to Reviewer 1, point 4.

Reviewer #3 (Remarks to the Author):

1, It is important to investigate whether the anti-androgen-mediated reduction in TMPRSS2 mRNA expression results in reduced SARS-CoV-2 entry. These analyses can be conducted fast with SARS-CoV-2 spike protein-pseudotyped vectors.

We have performed the suggested pseudotyped, and also live SARS-CoV-2, viral experiments and do see significant inhibition of viral entry/infection when cells are treated with bicalutamide and enzalutamide, supporting our hypothesis (Figure 7A and B). Thank you for the suggestion, which we agree very much strengthens the impact of the study.

2, Figure 1 is not convincing. The cell lines should be treated using the same protocol, including the same androgen.

As suggested, Figure 1 has been repeated using DHT used (and a second antiandrogen, bicalutamide, included).

3, “Coronaviruses have structural (Spike, Nucleocapsid, Matrix and Envelope)”. “Matrix” should read “Membrane”

Thank you. This has been corrected.

4, “TMPRSS2 primes the viral Spike (or S) protein by cleaving it at two sites; this facilitates fusion of the viral and host membranes [12-14].” This statement is incorrect – S1/S2 is cleaved by furin and S2 is cleaved by TMPRSS2 – and the references are outdated. Please consider citing PMID: 32376634 and PMID: 32362314.

We have added these references and modified the text (Page 2). “Cellular entry of SARS-CoV-2 requires host proteins expressed on the epithelial cell surface, most essential are Transmembrane Serine Protease 2 (TMPRSS2) and Angiotensin-Converting Enzyme 2 (ACE2)¹¹⁻¹³. TMPRSS2 cleaves and primes the viral Spike (or S) protein; this facilitates fusion of the viral and host membranes¹⁴⁻¹⁸.”

6, “Cellular entry is subsequently facilitated by ACE2, “. In the view of this reviewer, it has not been demonstrated that SARS-CoV-2 spike is first cleaved and then binds to ACE2. For SARS-CoV-1, the opposite has been suggested.

Thank you, we have removed the word “subsequently” to avoid confusion (Page 2). “Cellular entry is facilitated by ACE2, a terminal carboxypeptidase and type I transmembrane glycoprotein¹⁹.”

REVIEWER COMMENTS

Reviewer #1 (Remarks to the Author):

The authors made a significant improvement for this study. Some of the new data are very impressive. There are still some concerns that need to be addressed:

1. In Fig. 1, 10nM DHT should induce TMPRSS2 ~5-10 fold in LNCaP cells. However, only ~1.6 fold increase was shown in this figure.
2. In Fig. 4, DHT-induced TMPRSS2 expression in H1944 cells. The protein expression of AR and TMPRSS2 should be measured in this model as well.
3. In Fig. 6C, the ChIP needs to be shown in replicates.
4. Chinnaiyan group has recently published a similar work in PNAS (PMID:33310900). This should be cited and discussed. In particular, they show that Ciliated cells (bronchial) have very low AR and AT2 cells have very low AR and TMPRSS2.

Reviewer #2 (Remarks to the Author):

The authors have added several pieces of new data. However, a majority of the reviewer's concerns remain as detailed below.

1. The manuscript remained very limited in its scope, novel findings, and impacts. It may be more appropriate for publication in a specialized journal.
2. The title is an over statement. There is no data in the manuscript direct supporting Enzalutamide reduces cellular entry of SARS-COV-2.
3. The novelty of the study is greatly lacking. AR and TMPRSS2 involvement in COVID-19 has been previously reported in multiple studies. Therefore, Figures 1 to 5 are confirmatory.
4. Figure 6 identified a novel enhancer at 100-200kb upstream of TMPRSS2 gene that is specific to lung tissues. However, there is no data to prove that this enhancer indeed regulates TMPRSS2. The authors showed AR binding to this region using ChIP-qPCR, but this does not support that AR regulates TMPRSS2, rather than several other genes in this region.
5. The manuscript does not provide a therapeutic option, developed specifically from their own novel findings, for COVID-19. For example, is the novel enhancer turned on by a specific mechanism in lung tissues that can be targeted using a particular drug as a treatment for COVID-19? At the least, experimental data to show the effects of enhancer and its inhibition on viral entry should be provided using in vitro and in animal models.

Reviewer #3 (Remarks to the Author):

The authors have adequately addressed the comments raised by this reviewer and the findings reported are of significant interest. However, some, mostly minor points, remain to be addressed.

"The COVID-19 pandemic, caused by the novel human coronavirus SARS-CoV-2 coronavirus, attacks various organs but most destructively the lung." Formally, the disease COVID-19 (and not the pandemic) attacks the lung.

"we assessed uptake of SARS-CoV-2". Uptake of virus particles into cells does not necessarily lead to infection. It is suggested to replace "uptake" by "entry".

"2002 SARS pandemic" should read "2002/2003 SARS pandemic"

The results shown in figure 7A should be interpreted more carefully. Thus, it has not been examined whether the effects were SARS-CoV-2 spike protein specific, i.e. were not observed with a control glycoprotein like VSV-G, and were TMPRSS2-specific, i.e. were not observed upon directed expression of TMPRSS2.

We would like to thank the reviewers for taking time to carefully review our manuscript and for the constructive feedback provided. We have responded to the comments below:

Reviewer #1 (Remarks to the Author):

The authors made a significant improvement for this study. Some of the new data are very impressive. There are still some concerns that need to be addressed:

1. In Fig. 1, 10nM DHT should induce TMPRSS2 ~5-10 fold in LNCaP cells. However, only ~1.6 fold increase was shown in this figure.

The reviewer is correct that more robust induction is often reported - this varies according to hormone concentration and time of incubation. We have repeated the assay and find that TMPRSS2 expression is induced approximately 5-fold at the 6hr time point. The text (page 3) and figure (Figure 1A) have been updated.

2. In Fig. 4, DHT-induced TMPRSS2 expression in H1944 cells. The protein expression of AR and TMPRSS2 should be measured in this model as well.

Protein expression analysis of AR and TMPRSS2, in response to DHT and enzalutamide, has been added to Figure 4C. As expected, AR and TMPRSS2 levels increase in response to DHT and this is reversed following enzalutamide treatment. The text (page 4) and figure (Figure 4c) have been updated.

3. In Fig. 6C, the ChIP needs to be shown in replicates.

Replicates have now been included and the figure (Figure 6C) and text (page 6) modified.

4. Chinnaiyan group has recently published a similar work in PNAS (PMID:33310900). This should be cited and discussed. In particular, they show that Ciliated cells (bronchial) have very low AR and AT2 cells have very low AR and TMPRSS2.

This paper, which was published after we had re-submitted the manuscript, has now been cited and we have discussed some of the findings in relation to the data presented here (page 3 and page 6). Importantly, the findings of the Chinnaiyan group fully support the data presented here, for example they state:

Qiao et al. 2021, Proceedings of the National Academy of Sciences Jan 2021, 118 (1) e2021450118; DOI: 10.1073/pnas.2021450118

"The results demonstrated that AR was expressed with TMPRSS2 and ACE2 in several types of human (Fig. 1A) and murine (Fig. 1 B and C and SI Appendix, Fig. S2) lung epithelial cells, including alveolar (AT1 and AT2) epithelial cells, with maximal expression in ciliated and secretory epithelial cells (bronchial cells). This suggested that pulmonary TMPRSS2 and ACE2 in alveolar and bronchial cells had the potential to be regulated by AR."

Reviewer #2 (Remarks to the Author):

The authors have added several pieces of new data. However, a majority of the reviewer's concerns remain as detailed below.

2. The title is an over statement. There is no data in the manuscript direct supporting Enzalutamide reduces cellular entry of SARS-COV-2.

Respectfully, we have indeed shown data supporting that enzalutamide reduces cellular entry of SARS-COV-2. In Figure 7A, we used the pseudotyped virus system to demonstrate that enzalutamide, and another antiandrogen bicalutamide, significantly reduce cellular entry. Further, in Figure 7C we show that the antiandrogens successfully inhibit SARS-CoV-2 (live virus) infection.

1. The manuscript remained very limited in its scope, novel findings, and impacts. It may be more appropriate for publication in a specialized journal.

3. The novelty of the study is greatly lacking. AR and TMPRSS2 involvement in COVID-19 has been previously reported in multiple studies. Therefore, Figures 1 to 5 are confirmatory.

This is the first study to show unequivocal evidence that antiandrogens, acting via the androgen receptor, reduce SARS-CoV-2 infection of human lung cells. The logical and well-supported implications - that widely prescribed and well-tolerated antiandrogen therapies could be used to combat the COVID-19 pandemic and possible other respiratory viruses – have enormous impact and mainstream interest.

4. Figure 6 identified a novel enhancer at 100-200kb upstream of TMPRSS2 gene that is specific to lung tissues. However, there is no data to prove that this enhancer indeed regulates TMPRSS2. The authors showed AR binding to this region using ChIP-qPCR, but this does not support that AR regulates TMPRSS2, rather than several other genes in this region.

5. The manuscript does not provide a therapeutic option, developed specifically from their own novel findings, for COVID-19. For example, is the novel enhancer turned on by a specific mechanism in lung tissues that can be targeted using a particular drug as a treatment for COVID-19? At the least, experimental data to show the effects of enhancer and its inhibition on viral entry should be provided using in vitro and in animal models

We have identified a region with the potential to be a novel enhancer of TMPRSS2 expression in lung and have demonstrated that AR preferentially binds to this region in lung cells, in contrast to the binding seen in prostate cells. This is suggestive of a potential tissue-specific regulation, but we accept that further experiments would be needed to prove whether this is the case. We have added text on page 6 to clarify this.

The second point is beyond the scope of this study, but whether or not any novel enhancer may have a specific mechanism that could be targeted by a particular drug, here we have shown that bicalutamide and enzalutamide reduce TMPRSS2 expression and that these antiandrogens block viral entry (pseudotyped virus) and infection (live virus) in lung cells. We therefore propose antiandrogens as a therapeutic option for COVID-19. In addition to the lung, it has become increasingly apparent that SARS-CoV-2 targets other organs including skin, the cardiovascular system, kidneys, and the male genital tract. We therefore believe that down-regulation of TMPRSS2 expression in multiple tissues, through the use of antiandrogens, will be more effective than therapies that specifically target TMPRSS2 expression in the lung.

Reviewer #3 (Remarks to the Author):

The authors have adequately addressed the comments raised by this reviewer and the findings reported are of significant interest. However, some, mostly minor points, remain to be addressed.

1, “The COVID-19 pandemic, caused by the novel human coronavirus SARS-CoV-2 coronavirus, attacks various organs but most destructively the lung.” Formally, the disease COVID-19 (and not the pandemic) attacks the lung.

Apologies for that error, this has been changed to:

COVID-19, caused by the novel human coronavirus SARS-CoV-2 coronavirus, attacks various organs but most destructively the lung.

2, “we assessed uptake of SARS-CoV-2”. Uptake of virus particles into cells does not necessarily lead to infection. It is suggested to replace “uptake” by “entry”.

This has been changed to “entry” (Abstract, page 1)

“2002 SARS pandemic” should read “2002/2003 SARS pandemic”

This has been changed to 2002/2003 (page 2)

The results shown in figure 7A should be interpreted more carefully. Thus, it has not been examined whether the effects were SARS-CoV-2 spike protein specific, i.e. were not observed with a control glycoprotein like VSV-G, and were TMPRSS2-specific, i.e. were not observed upon directed expression of TMPRSS2.

We have repeated the pseudotype experiment and confirmed that TMPRSS2 expression is down-regulated by enzalutamide at the point of transduction (Supplemental Figure 3a-c). We have also performed the VSV-G experiment (Figure 7B) and demonstrated that entry is not affected by antiandrogens. This confirms that antiandrogens have specificity for spike protein-directed viral entry and significantly strengthens the manuscript – we thank the reviewer for this suggestion. We have also responded to their comment about interpreting the results more carefully and the text has also been modified to state that the antiandrogens may inhibit viral entry via downregulation of factors in addition to TMPRSS2 (pages 6-7).

REVIEWERS' COMMENTS

Reviewer #1 (Remarks to the Author):

No further comments.

Reviewer #2 (Remarks to the Author):

Unfortunately, my concerns remain unaddressed. The manuscript has very little novelty and impact, considering the effects of androgen on SARS-CoV-2 is well known to the field, supported by many publications (e.g. Samuel, 2020 Cell Stem Cell; Qiao, 2020, PNAS; Li 2021 Nature Communications; Deng, 2021 iScience) and clinical trials already ongoing for a while. The data presented in this manuscript is much thinner than what's already published.

Reviewer #3 (Remarks to the Author):

The authors have adequately addressed the points raised by this reviewer. Two minor issues remain and can probably be fixed at the proofing stage:

It is unclear why VSV-G bearing particles entered VC treated cells with roughly 80% efficiency (Figure 7A) - how were the data normalized?

"live virus" is lab jargon and should be avoided

Reviewer #1 (Remarks to the Author):

No further comments.

Reviewer #2 (Remarks to the Author):

Unfortunately, my concerns remain unaddressed. The manuscript has very little novelty and impact, considering the effects of androgen on SARS-CoV-2 is well known to the field, supported by many publications (e.g. Samuel, 2020 Cell Stem Cell; Qiao, 2020, PNAS; Li 2021 Nature Communications; Deng, 2021 iScience) and clinical trials already ongoing for a while. The data presented in this manuscript is much thinner than what's already published.

We have addressed the editorial concerns relating to this and appreciate the stance of Nature Communication not to penalise papers on the basis of novelty when competing papers appear during the review process.

Reviewer #3 (Remarks to the Author):

The authors have adequately addressed the points raised by this reviewer. Two minor issues remain and can probably be fixed at the proofing stage:

It is unclear why VSV-G bearing particles entered VC treated cells with roughly 80% efficiency (Figure 7A) - how were the data normalized?

Figure 7A has been replotted so that the data are normalised to the VC treatment. The VC treatment is now set at 100%.

"live virus" is lab jargon and should be avoided

All usage of the term "live virus" has been removed. The text has been changed as follows:

In the abstract, which has been shortened to comply with the word limit, reference to "live virus" has been removed.

Page 6 – text changed from "To see if antiandrogens could inhibit SARS-CoV-2 infection, TCID₅₀ assays were performed with the live virus (Figure 7B)." to "To see if antiandrogens could inhibit SARS-CoV-2 infection, TCID₅₀ assays were performed with the virus (Figure 7B)."

Page 6 – text changed from "However, antiandrogens were effective in the live virus experiments, in cells with exogenous ACE2 over-expressed..." to "However, antiandrogens were effective in the SARS-CoV-2 experiments, in cells with exogenous ACE2 over-expressed..."

Page 10 - In the methods, the sub-heading "SARS-CoV-2 Live Infection Studies" has been changed to "SARS-CoV-2 Infection Studies"

Figure 7B has been changed from "Live Virus" to "SARS-CoV-2"